# Blood Cultures Time-to-Positivity as an Antibiotic Stewardship Tool in Immunocompromised Children with Gram-Negative Bacteraemia

**DOI:** 10.3390/antibiotics14080847

**Published:** 2025-08-21

**Authors:** Julià Gotzens, Aina Colom-Balañà, Manuel Monsonís, Laia Alsina, María Antonia Ruiz-Cobo, María Ríos-Barnés, Anna Gamell, Eneritz Velasco-Arnaiz, Irene Martínez-de-Albéniz, Victoria Fumadó, Clàudia Fortuny, Antoni Noguera-Julian, Sílvia Simó-Nebot

**Affiliations:** 1Infectious Diseases and Systemic Inflammatory Response in Pediatrics, Pediatric Infectious Diseases Department, Institut de Recerca Sant Joan de Déu, Hospital Sant Joan de Déu, 08950 Esplugues de Llobregat, Spain; julia.gotzens@sjd.es (J.G.); maria.rios@sjd.es (M.R.-B.); annamaria.gamell@sjd.es (A.G.); eneritz.velasco@sjd.es (E.V.-A.); irene.martinezd@sjd.es (I.M.-d.-A.); victoria.fumado@sjd.es (V.F.); claudia.fortuny@sjd.es (C.F.); 2Department of Surgery and Medical-Surgical Specialties, Faculty of Medicine and Health Sciences, Universitat de Barcelona (UB), 08036 Barcelona, Spain; laia.alsina@sjd.es; 3Pediatrics Department, Hospital Sant Joan de Déu, 08950 Esplugues de Llobregat, Spain; aina.colomi@sjd.es; 4Microbiology Department, Hospital Sant Joan de Déu Barcelona, 08950 Esplugues de Llobregat, Spain; manuel.monsonisc@sjd.es; 5Clinical Immunology and Primary Immunodeficiencies Unit, Pediatric Allergy and Clinical Immunology Department, Hospital Sant Joan de Déu, 08950 Esplugues de Llobregat, Spain; 6Leukemia, Lymphoma and CART Department, Pediatric Cancer Center Barcelona (PCCB), Hospital Sant Joan de Déu de Barcelona, 08950 Esplugues de Llobregat, Spain; mariaantonia.ruiz@sjd.es; 7Center for Biomedical Network Research on Epidemiology and Public Health (CIBERESP), 28029 Madrid, Spain; 8Translational Research Network in Pediatric Infectious Diseases (RITIP), 28029 Madrid, Spain

**Keywords:** antimicrobial stewardship, bloodstream infection, febrile neutropenia, Gram-negative, immunocompromised, oncology, time-to-positivity

## Abstract

**Background/Objectives**: Children and adolescents with haematologic malignancies or other causes of immunosuppression are at high risk of severe infections. Determining the probability of Gram-negative bacilli bloodstream infections (GNB-BSI) within 24 h of blood culture (BC) incubation could support early antibiotic de-escalation, compared to the current guidelines recommending de-escalation after 48–72 h. **Methods**: Retrospective, observational single-centre study describing BC time-to-positivity (TTP) in GNB-BSI in a paediatric cohort of immunocompromised children. **Results**: In 128 episodes (100 patients), TTP was less than 24 h in >95% cases. TTP did not differ based on sex, underlying disease, degree of neutropenia, or PICU admission. Antibiotic initiation prior to BC collection and microbiological aetiology (microbiological aetiology different from *Pseudomonas aeruginosa*, *Escherichia coli*, *Klebsiella pneumoniae*, or *Enterobacter cloacae*) were the only identified risk factors associated with BC growth beyond 24 h. No patients with late BC growth died or required PICU admission. **Conclusions**: If BC remains negative after 24 h of incubation, GNB-BSI is unlikely in immunocompromised children and adolescents with fever. These results support early de-escalation strategies, shortening unnecessary exposure to broader-spectrum antibiotics, and potentially decreasing adverse events and costs.

## 1. Introduction

Children and adolescents with haematologic malignancies or other causes of immunosuppression are at high risk of severe infections, including bloodstream infections (BSI). Approximately 20% of children with onco-haematological diseases develop a BSI during treatment, with 40−50% of these being caused by Gram-negative bacteria [1,2]. Mortality in paediatric immunocompromised patients with BSI is high (around 6.5%), especially during severe neutropenia. After the underlying disease itself, infections are the main cause of death in these patients [3,4].

Multidrug-resistant (MDR) infections are increasing in Europe, as demonstrated by multiple surveillance studies and systematic reviews. High and rising percentages for key pathogens have been reported, including carbapenem-resistant *Klebsiella pneumoniae* and *Acinetobacter baumannii*, with MDR organisms frequently involved both in community and healthcare-associated infections in all populations, including cancer patients [5,6]. Longitudinal surveillance shows increasing prevalence of extended-spectrum β-lactamase (ESBL) and carbapenemase-producing *Enterobacterales*, with notable geographic variation but a clear trend toward higher rates over time [7].

Early empirical broad-spectrum antibiotic treatment and in-hospital support have significantly decreased the mortality of paediatric patients with cancer and febrile neutropenia (FN) and sepsis [8]. Current guidelines recommend first-line therapy with a beta-lactam active against *Pseudomonas aeruginosa* for most patients with FN [9]. In critical patients or in patients at risk of MDR Gram-negative bacilli (GNB) infections (i.e., recent infection or colonization by MDR), the choice of the best empirical treatment is challenging, and the addition of a second agent with activity against GNB (i.e., aminoglycoside) or the use of carbapenems is recommended [4]. Unfortunately, this last scenario is becoming more frequent, as an increase in infections due to MDR-GNB has been detected in paediatric patients with cancer [10]. These antibiotic regimens are associated with important adverse events, such as nephro- and ototoxicity with aminoglycosides, neurotoxicity and seizures with carbapenems, or *Clostridioides difficile* infection and gut microbiota disruption with broad-spectrum β-lactams. They also promote the selection of MDR microorganisms and substantially increase treatment costs [4,10,11]. Strategies to quickly rule out GNB-BSI are needed to enable clinicians to safely de-escalate to standard antibiotic coverage.

Antimicrobial stewardship programs (ASPs) are effective strategies for reducing antimicrobial overuse and improving clinical outcomes across various settings [12]. Multiple studies demonstrate that ASPs in oncology and haematology wards reduce unnecessary antibiotic use, decrease the incidence of infections caused by MDR organisms, and do not increase adverse outcomes such as mortality or length of hospitalization [13,14]. Also, recent data suggest that paediatric ASPs can similarly reduce the use of antimicrobials such as carbapenems in cancer patients. However, data on the implementation and impact of ASPs in paediatric oncology wards remain limited [15,16]. Key challenges persist, particularly regarding risk stratification for BSI, antimicrobial development (paediatric formulations), and the optimal timing of antimicrobial de-escalation and duration of treatment of FN episodes [17].

Despite notable differences in physiology, immune response, and pharmacokinetics, paediatric protocols are often extrapolated from adult guidelines [18]. This practice is partly due to the relative scarcity of paediatric-specific clinical trials and the inherent challenges of conducting research in vulnerable populations (only 10% of trials registered between 2000 and 2023 in the International Clinical Trials Registry Platform included children) [19]. Immunocompromised children present unique considerations: their drug metabolism and clearance differ significantly from those of adults; they may exhibit distinct patterns of bacterial colonization and infection; and the long-term consequences of antimicrobial toxicity or resistance can be more profound [20]. These differences highlight the urgent need for age-specific data to inform effective and safe antibiotic stewardship frameworks in paediatric populations.

Blood-culture (BC) time-to-positivity (TTP)—the interval from the start of incubation to a positive alarm—is a readily available and clinically informative metric [21]. Over the last decades, the widespread adoption of continuous-monitoring automated BC systems—such as BacT/ALERT—has reduced time to detection, standardized processes, and made this information more accessible. These platforms enable real-time detection of microbial growth by continuously tracking metabolic by-products, providing precise, timestamped alerts. This represents a significant improvement over earlier methods, in which Gram stains were performed manually twice a day to check for microbial growth [22].

Interest in TTP has grown substantially, as evidenced by a fourfold rise in related publications on this topic over the last 10 years [21]. In clinical practice, the most extended use of TTP is to diagnose catheter-related BSI (defined as a difference >2 h in the TTP between two BCs, drawn simultaneously from the central venous catheter and from a peripheral vein). It has also been studied as a prognostic factor or to identify the source of infection [21]. In adult cancer patients, TTP has been used as a tool to guide antimicrobial therapy, as this population combines high risk of invasive infections and high antimicrobial exposure [23,24]. Nevertheless, paediatric evidence remains limited, consisting primarily of small, single-centre retrospective cohorts. To date, only one study has specifically examined children with FN, reporting that 86% of GNB-BSI BC became positive within 24 h. However, this study included fewer than 20 events and was not specifically designed to evaluate TTP-guided antibiotic de-escalation [25]. Determining the probability of GNB-BSI at 24 h of BC incubation could support early antibiotic de-escalation strategies, as compared to de-escalation at 48−72 h, as recommended in current guidelines [4,9].

The primary objectives of this study were to characterise the distribution of BC-TTP among immunocompromised children and adolescents with GNB-BSI and to investigate patient-, clinical-, and pathogen-related factors that influence TTP, with the ultimate goal of informing evidence-based, early antibiotic de-escalation strategies in this high-risk population.

## 2. Results

### 2.1. Study Population

During the study period, 147 episodes of GNB-BSI were identified, of which 19 (12.9%) were excluded due to different reasons, including polymicrobial bacteriemia (n = 13), patient’s age > 18 years at diagnosis (n = 3), lack of data about TTP (n = 1), and contaminant isolates (n = 2). Overall, 128 episodes of GNB-BSI were identified in 100 patients (males n = 63, 63.0%), with mean (SD) age at diagnosis being 7.7 (4.9) years.

The most common underlying diseases were leukaemia (n = 50, 50.0%) and solid tumours (n = 43, 43.0%). A total of 20 GNB-BSI episodes (15.6%) occurred after hematopoietic stem cell transplant (HSCT)—allogenic HSCT (n = 17) and autologous HSCT (n = 3)— and 2 (1.6%) during chimeric antigen receptor T-cell (CAR-T) therapy (day 1 and day 20 after infusion, respectively).

The median (IQR) absolute neutrophil count (ANC) at the onset of GNB-BSI was 0/mm^3^ (0–200/mm^3^), and severe neutropenia (<500 neutrophils/mm^3^) was present at diagnosis in 99 episodes (77.3%). The median (IQR) peak CRP level during the first 48 h was 107 mg/L (59–177 mg/L). Procalcitonin (PCT) was determined at least once within the first 48 h in 106 episodes (82.8%), with a median (IQR) peak value of 1.52 ng/mL (0.49–6.8 ng/mL).

### 2.2. Microbiological Findings

The most frequently isolated microorganisms were *Escherichia coli* (n = 41, 33.0%), *Pseudomonas aeruginosa* (n = 25, 19.5%), and *Klebsiella pneumoniae* (n = 22, 17.2%). Other bacteria included *Enterobacter cloacae* complex (n = 13, 10.3%), *Klebsiella oxytoca* (n = 6, 4.7%), *Klebsiella aerogenes*, *Klebsiella variicola*, and *Acinetobacter baumannii* (n = 2 each, 1.6%). Bacteria expressing acquired antibiotic resistance mechanisms were detected in 21 episodes (16.4%), with ESBL (n = 13, 10.1%) and carbapenemases (n = 6, 4.7%) being the most frequent. Carbapenemase types were VIM (n = 3), NDM (n = 2), and OXA-48 (n = 1).

Overall, the isolated strains demonstrated in vitro susceptibility to piperacillin-tazobactam in 99 cases (77.3%), amikacin in 117 cases (91.4%), and meropenem in 121 cases (94.5%).

### 2.3. Empirical Therapy and Clinical Outcomes

Carbapenems or dual therapy (antipseudomonal non-carbapenem β-lactam combined with an aminoglycoside) were used in 96 episodes (75.0%) as empirical treatments, with each regimen employed in 48 episodes (37.5%). In 7 episodes (5.4%), the isolated strains were resistant to empirical antibiotic therapy, including ESBL-producing *Escherichia coli* (n = 2), *Campylobacter jejuni* (n = 1), *Acinetobacter lwoffii* (n = 1), *Stenotrophomonas maltophilia* (n = 1), OXA-10–producing *E. coli* (n = 1), and NDM-carbapenemase–producing *Klebsiella oxytoca* (n = 1). The median (IQR) duration of antibiotic therapy was 10 days (8–11 days), and median length of hospitalization was 21 days (11–42 days). In 20 episodes (15.6%), antibiotic with GNB coverage was already in use at the time of BC collection.

A total of 15 patients required admission to the PICU (11.7%) (Appendix A), and 7 patients died within 30 days after bacteraemia (5.5%), with GNB-BSI being the cause of death in 2 of them (Appendix A). In total, 20 patients (15.6%) experienced the composite outcome of PICU admission or 30-day mortality. The two episodes in which the patients died involved a 4-year-old child with relapsed acute lymphoblastic leukaemia receiving palliative care, who developed a BSI due to *E. coli*, and a 3-month-old infant with a renal tumour (congenital mesoblastic nephroma) who developed a fatal urinary septic shock caused by *E. cloacae*.

### 2.4. BC-TTP Distribution and Associated Clinical and Microbiological Factors

The median (IQR) BC-TTP was 11.3 (9.4–13.6) hours. TTP was <24 and <36 h in 95.3% (122/128) and 97.7% (125/128) of cases, respectively. Six episodes of GNB-BSI (4.7%) had TTP > 24 h (clinical and microbiological characteristics shown in Table 1); none of these patients required PICU admission or died within 30 d of bacteraemia. No cases of MDR GNB-BSI were detected after 24 h of incubation.

BC positivity results related to time are shown in Figure 1. No differences were detected in TTP according to underlying disease, severity of neutropenia at diagnosis, foci of bacteraemia, need for PICU admission, or death at day 30 (Table 2). The median TTP for *Pseudomonas aeruginosa* BSI was longer than for other GNB (16.0 vs. 10.8 h, respectively; *p* 0.041). Shorter TTP showed a trend toward association with worse outcomes (30-day mortality and/or PICU admission), although this did not reach statistical significance (OR 0.860, 95% CI 0.734–1.008; *p* = 0.062).

### 2.5. Factors Associated with TTP > 24 h

When comparing episodes with TTP > 24 h versus < 24 h, aetiology different from *Pseudomonas aeruginosa*, *Escherichia coli*, *Klebsiella pneumoniae*, or *Enterobacter cloacae* (5/6 cases, 83.3% vs. 23/122, 18.9%, *p* = 0.005) and previous antibiotic treatment (3/6 cases, 50.0% vs. 17/122 cases, 13.9%, *p* = 0.048) were more common in the former. No other significant differences were observed between the groups (Table 3).

## 3. Discussion

We report a large case series describing BC-TTP in paediatric immunocompromised patients with GNB-BSI, predominantly comprising patients with cancer and severe neutropenia. TTP was <24 h from incubation in 95.3% of cases, irrespective of sex, underlying disease, severity of neutropenia, focus of infection, need for admission to the PICU, or 30-day mortality. Although our results should be interpreted with caution, due to the observational design and the small number of patients with TTP > 24 h, these findings support early antibiotic de-escalation from carbapenems or dual regimens to other beta-lactams monotherapy at 24 h of BC incubation in clinically stable patients, provided that the BC is negative, rather than at 48–72 h, as indicated in many guidelines [4,9]. After 24 h of BC incubation without growth, GNB-BSI would be infrequent and less severe. For this strategy to be feasible, microbiological results must be accessible around the clock, and real-time communication between infectious disease and microbiology teams is essential.

Our results are similar to those previously reported in immunocompetent children. A study that evaluated 1454 paediatric-positive BC, including 341 episodes of GNB bacteraemia, reported 88.0% of GNB-BSI cultures to grow within 24 h (median [IQR] TTP: 14.3 [11.6–18.3] h) [26]. In another study including 562 episodes of bacteraemia and 185 GNB-BSI, TTP was <24 h in 88.0% of BC and raised to 91.0% in GNB-BSI [27]. Finally, a cohort study detailing the aetiology of 377 episodes of FN in paediatric oncology patients reported that 13 out of 15 (86.6%) cases of GNB-BSI had a TTP below 24 h, with the two exceptions being caused by *Capnocytophaga sputigena*, a slow-growing bacillus [25].

Similar findings have been reported in adults with cancer and FN. A prospective study conducted over a 14-year period evaluated 465 episodes of GNB-BSI and reported BC-TTP to be below 24 h in 96.5% of cases [23]. Another study evaluated 129 episodes of BSI and reported that 48 out of 49 (98.0%) GNB-BSI episodes had a TTP < 24 h [24]. These studies also support the early reassessment of empirical therapy—potentially as early as 24 h—if BC remain negative [23,24].

BC-TTP has mainly been associated with the microbiological aetiology. GNB grow faster than Gram-positive bacteria and fungi [24,26]. Within GNB, enterobacteria grow faster than *Pseudomonas aeruginosa* and strict anaerobes [28]. This is consistent with our findings, where the TTP for BSI due to *Pseudomonas aeruginosa* was significantly longer as compared to other GNB (mostly enterobacteria), although it remained < 24 h in all cases.

Other than the intrinsic growth kinetics of pathogens, BC-TTP is thought to be influenced also by pathogen load in the blood sample, host-specific factors, and laboratory-processing variables [29]. Despite the use of resins to neutralize antimicrobial agents in BC bottles, antibiotic treatment is a common cause of sterile BC [22]. Sterilization of BC depends on antibiotic concentration at the time of sample collection (more bacterial recovery during trough concentrations) and type of antibiotics (resins are less active against carbapenems and fluoroquinolones) [22]. In our study, no differences in TTP were observed between patients with GNB-BSI that had received antibiotics before BC collection and those who had not (TTP 11.5 vs. 11.3 h, *p* = 0.257), but previous exposure to antibiotics was associated with TTP > 24 h (13.9% vs. 50.0%, *p* = 0.048).

It is important to highlight that the growth dynamics—specifically, the short TTP of GNB-BSI caused by *Enterobacterales*—is critical to the safe implementation of our therapeutic strategy. This is particularly relevant given that ESBL-producing and carbapenem-resistant pathogens—both predominantly associated with *Enterobacterales*—represent the highest estimated burden among MDR-GNB bacteria due to their widespread prevalence [30]. These pathogens are precisely the targets of broad-spectrum antimicrobial regimens in FN, including carbapenems or the combination of a non-carbapenem beta-lactam with an aminoglycoside. In our study, no MDR-GNB pathogens were isolated after 24 h of incubation. Similarly, Puerta et al. reported no growth of multidrug- or extensively drug-resistant organisms beyond 24 h in adult patients with FN [22].

Dierig et al. did not identify differences in TTP according to sex, age, focus, or severity of infection in 562 previously healthy children with sepsis [27]. Similarly, MacBryne et al. found no differences in TTP according to age, focus of infection, or treating department in over 1400 positive paediatric BC [26]. Our results did not show these factors or the severity of neutropenia influenced BC-TTP.

TTP has also been studied as a prognostic tool because of its potential relationship with bacterial load [31,32,33]. The shorter the TTP is, the higher the bacterial load would be, potentially leading to worse outcomes. In adults, the role of TTP as an independent predictor of mortality is well-documented, with each hour of shorter TTP in GNB-BSI estimated to increase mortality by 10% [31]. In children, two studies analysing 52 and 94 episodes of *Pseudomonas aeruginosa* and *Klebsiella pneumoniae* bacteraemia, respectively, also observed that a shorter TTP was linked to worse outcomes, including septic shock and in-hospital mortality [31,32]. In our study, we found no significant differences in BC-TTP between patients who required PICU admission and those who did not (*p* = 0.175), although there was a trend toward an association between shorter TTP and worse outcomes, including 30-day mortality or PICU admission (*p* = 0.062).

In our series, antibiotic treatment prior to BC collection (*p* = 0.048) and the identification of certain GNB (*p* = 0.005) were both significantly associated with delayed BC growth (TTP > 24 h). Out of six such cases, half of these episodes were CRBSI, a condition previously associated with prolonged TTP in adults [23,24]. Also, GNB known to grow slowly—such as *Campylobacter* and non-*aeruginosa Pseudomonas* spp.—were identified in two patients [27]. The remaining episode (*Acinetobacter* spp., TTP 25.7 h) was the only case in which early antibiotic de-escalation at 24 h would have been inappropriate based on in vitro susceptibility testing. No patients with TTP > 24 h died or required PICU admission. It can be hypothesized that GNB-BSI with TTP > 24 h are less aggressive due to less virulent pathogens (i.e., slow-growing pathogens), lower bacterial loads, or antibiotics are initiated prior to BC collection, as observed in our cohort.

Based on our findings regarding BC-TTP, we propose a clinical algorithm (Figure 2) including early de-escalation of antibiotic therapy in immunocompromised children with fever at risk of GNB-BSI. It should be kept in mind that this preliminary proposal is exploratory in nature and requires validation in external cohorts, ideally through larger, prospective, multicentre studies. In this model, empirical therapy—tailored to the risk of MDR infections at presentation—should be maintained for at least 24 h. If BC becomes positive within that time frame, antimicrobial therapy should be adjusted according to the identified pathogen and its susceptibility profile. Conversely, if BC remains negative at 24 h and the patient is clinically stable, de-escalation to narrower-spectrum agents could be safely considered.

This strategy aligns with antimicrobial stewardship principles, balancing patient safety and the goal of minimizing unnecessary antibiotic exposure. Importantly, this approach should be adapted to local context, taking into account laboratory turnaround times, ASP capacity, and local epidemiology of pathogens and resistance. In our view, patients whose BC were obtained after antibiotic initiation may require an extended observation period before considering de-escalation.

As previously stated, the retrospective observational design and the relatively small sample size are evident limitations of our study. We were unable to document the BC volume, which may have introduced variability—particularly due to the common challenges in obtaining adequate venous access in pediatric patients and the wide range of patient weights (from infants to adolescents). This heterogeneity may have led to differences in blood volume processing, potentially influencing TTP measurements. We were unable to capture the time interval between sample collection and incubation in the Microbiology Laboratory, which led to an underestimation of the BC-TTP that might have been clinically significant [34]. Additional limitations arise from the single-centre nature of this study, conducted in a high-income setting where broad-spectrum antibiotic prophylaxis is not routinely used for patients with prolonged neutropenia. In contrast, settings that implement fluoroquinolone prophylaxis and experience outbreak-driven BSI may observe different TTP dynamics. Furthermore, regional variations in bacterial ecology—particularly a higher prevalence of non-fermenting GNB or strict anaerobes—could further influence TTP measurements and limit the generalizability of our findings.

## 4. Materials and Methods

### 4.1. Study Design and Setting

We performed a retrospective observational single-centre cohort study in a tertiary paediatric hospital in Barcelona (Spain), including a dedicated centre for paediatric oncology with a capacity of 37 inpatient beds, 8 HSCT rooms, and 26 hospital day care units, where over 1100 admissions of immunocompromised paediatric patients are managed every year.

The ASP at our institution is based on post-prescription review with feedback and was implemented in January 2017 [12]. Daily reviews of antimicrobial prescriptions are conducted, and the specialist participates in daily ward rounds to discuss patients and optimize antimicrobial use. Although pre-prescription authorization is not required, prescription filters are in place for selected antimicrobial agents—such as carbapenems, linezolid, and some antifungals—requiring prescribers to specify the indication. The program also provides consultation support for outpatients, provides periodic feedback on antimicrobial consumption to prescribers, and contributes to the development and dissemination of clinical guidelines.

### 4.2. Study Population

Patients aged up to 18 years old with solid tumours, oncohaematologic conditions (leukaemia and lymphoma), HSCT recipients, CAR-T therapy, those affected with inborn errors of immunity and non-malignant haematological diseases, diagnosed with GNB-BSI during the period 2020–2023, were included.

Patients with polymicrobial infections (>1 microorganism isolated in the BC) and those in whom microbiological isolations were considered contaminants (commensal flora isolated in only 1 BC) were excluded [24].

### 4.3. Management of FN

According to our local FN protocol, at the onset of every febrile episode, two aerobic blood-culture sets are obtained before the first dose of empirical antibiotics: one from a peripheral vein and one from the central venous catheter (or from each lumen if multi-lumen). Anaerobic bottles are not collected routinely.

In our institution, surveillance cultures (rectal swabs) were performed periodically (weekly during admissions), and initial empirical treatment (at least for the first 24 h) was guided by these results. Systemic antibacterial prophylaxis was not routinely used in neutropenic patients, except for trimethoprim/sulfamethoxazole for the primary prophylaxis of *Pneumocystis jirovecii* infection.

Patients without surveillance cultures, those transferred from other hospitals, or those with positive surveillance cultures for an MDR organism within the past 3–6 months were considered at risk for MDR infections. Initiation of carbapenem, with or without a glycopeptide and with or without an aminoglycoside (depending on the suspected infectious focus), was standard of care for episodes presenting with septic shock.

### 4.4. Microbiological Procedures and Definitions

BC are processed using the BACT/ALERT 3D-automated microbial detection system (bioMérieux, Durham, NC, USA). When a BC flags positive, a Gram stain is immediately performed, followed by conventional subcultures on appropriate media (e.g., blood agar or other selective media, depending on the Gram stain result). Once bacterial growth is observed, microorganism identification is carried out using matrix-assisted laser desorption ionization–time of flight mass spectrometry (MALDI-TOF).

Antimicrobial susceptibility testing is performed using commercial panels, such as the *MicroScan* system (Beckman Coulter, Brea, CA, USA). Mechanisms of multidrug resistance are further characterized using multiplex lateral-flow immunochromatographic assays, including the *NG-Test CARBA 5* (NG Biotech, Guipry, France) for the detection of carbapenemases and the *NG-Test CTX-M MULTI* (NG Biotech, Guipry, France) for extended-spectrum β-lactamases (ESBLs).

BC-TTP was reported as the time elapsed from incubation to BC positivity alert. Information about the time between blood sampling and start of incubation and BC blood volumes was not available. At our institution, clinicians receive automated notifications of positive BC alerts around the clock, pending subsequent validation and definitive identification by the microbiologist. During regular working hours, the microbiology laboratory also directly contacts the Paediatric Infectious Disease team or the team responsible for the patient to inform the positive result. In patients with multiple BC collected simultaneously at the onset of the episode—such as those obtained from both a central line and peripheral venipuncture—the lowest TTP was recorded.

The foci of bacteraemia were classified as follows: catheter-related bloodstream infection, complicated abdominal focus (including intra-abdominal and perianal abscesses, intra-abdominal surgical conditions, and neutropenic typhlitis), urinary tract infection, skin and soft tissue infection, others, and unknown.

Previous antibiotic treatment was defined as treatment with ≥1 antimicrobial agent with GNB activity within the 24 h preceding the collection of the BC. This group includes a range of scenarios: selected patients who were receiving fluoroquinolone prophylaxis (not standard practice); patients who developed a new fever or clinical deterioration while already on antibiotic treatment; and cases where BC were collected after the initiation of antibiotics.

### 4.5. Data Collection and Outcomes

Clinical and microbiological variables were collected, including sex and age at GNB-BSI diagnosis; underlying disease (leukaemia, solid tumour, HSCT recipients, inborn errors of immunity, medullary aplasia, or others); absolute neutrophil count (ANC) at GNB-BSI diagnosis (cells/mm^3^); maximum C-reactive protein levels (quantitative immunoturbidimetry; normal values < 15 mg/L) and procalcitonin (quantitative immunoassay; normal values < 0.5 ng/mL) in the first 48 h after onset of symptoms; probable focus of BSI; type and duration of antibiotic treatment; need of admission to the PICU; isolated microorganism and its phenotypic susceptibility results; and BC-TTP. Also, 30-day follow-up mortality rates were collected.

### 4.6. Statistical Analysis

Data were analysed using the SPSS statistical software (IBM, version 29; Chicago, IL, USA), applying tests for comparison of quantitative data (Student’s *t*-test, analysis of variance) and qualitative data (chi-square test, contingency table, Fisher’s exact test), as appropriate. *p*-values < 0.05 were considered significant. We conducted a logistic regression analysis to evaluate the association between BC-TTP (modelled as a continuous variable) and the principal outcome (30-day mortality and/or PICU admission). 

### 4.7. Ethical Considerations

The study was approved by the local Ethics Committee [reference 32-20], which waived informed consent due to its retrospective nature and collection of anonymized data exclusively. The research was conducted following the principles of the Declaration of Helsinki and national and institutional standards.

## 5. Conclusions

In summary, TTP was below 24 h in the majority of BC from immunocompromised children with GNB-BSI in our study. The administration of antibiotics prior to BC collection and certain GNB pathogens were the only factors associated with delayed BC positivity (>24 h). These findings suggest that TTP may serve as a valuable clinical tool to guide early de-escalation of empirical broad-spectrum antibiotic therapy in immunocompromised children at risk of GNB-BSI. Such an approach could help reduce unnecessary antibiotic exposure, thereby lowering the risk of antimicrobial resistance, minimizing drug-related toxicity, and decreasing healthcare costs.

Our results also highlight the importance of optimizing BC practices—including obtaining samples prior to antibiotic administration and standardizing blood volume collection—to improve the reliability and interpretability of TTP.

Further prospective, multicentre studies, and interventional trials are warranted to validate these findings and to assess the safety and effectiveness of TTP-guided antibiotic de-escalation strategies and shortened treatment durations in this high-risk population.

## Figures and Tables

**Figure 1 antibiotics-14-00847-f001:**
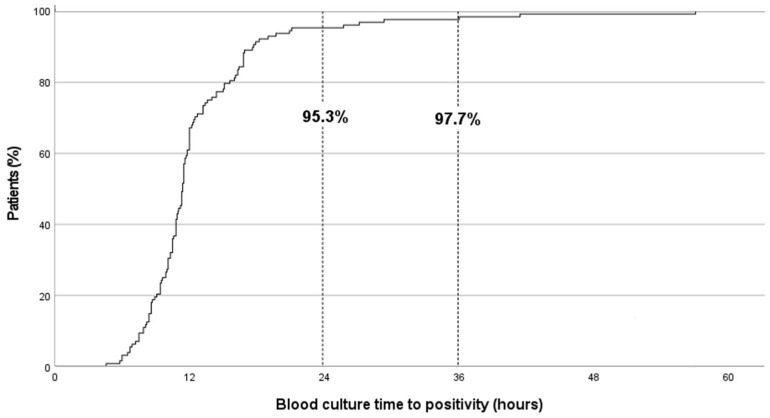
Blood-culture time-to-positivity results in paediatric immunocompromised patients (n = 128) with Gram-negative bacilli bloodstream infections.

**Figure 2 antibiotics-14-00847-f002:**
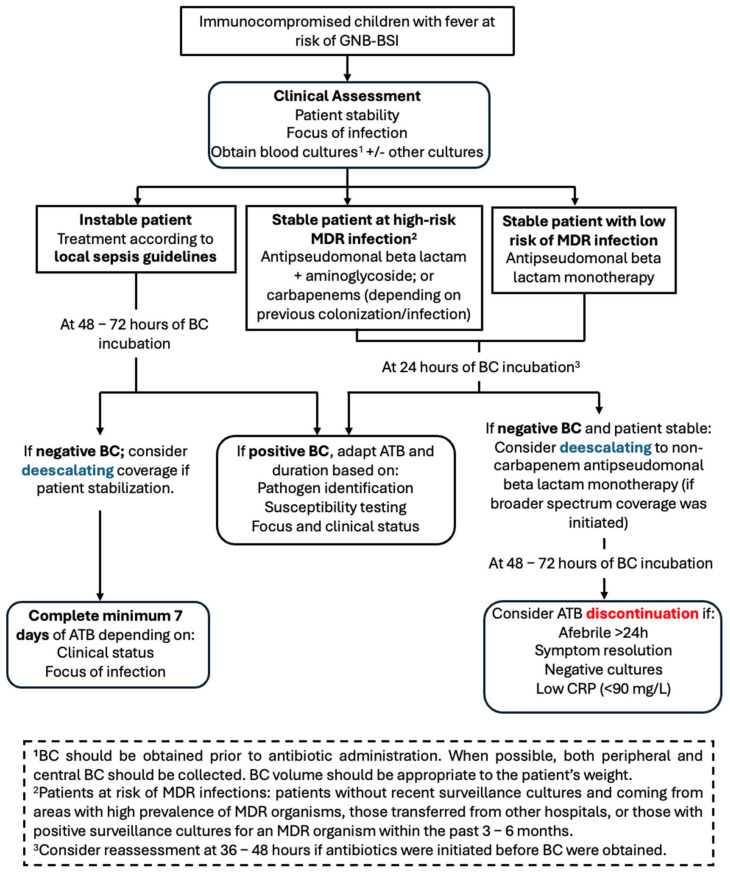
Proposal of clinical algorithm for early antibiotic de-escalation in paediatric patients at risk of GNB-BSI. ATB: antibiotics; BC: blood cultures; CRP: C-reactive protein; GNB-BSI: Gram-negative bacilli bloodstream infection; MDR: multidrug-resistant.

**Table 1 antibiotics-14-00847-t001:** Characteristics of Gram-negative bacilli bloodstream infections (GNB-BSI) in paediatric immunocompromised patients with blood cultures time-to-positivity (TTP) > 24 h.

TTP (h)	Sex/Age	Microorganism	Underlying Diseases	Infectious Focus	Antibiotic Pretreatment	ANC (/mm^3^)
25.7	Male/10 y	*Acinetobacter* spp.	HR-ALL, C/I	Unknown	No	0
27.1	Male/17 y	*Klebsiella pneumoniae*	Astrocytoma	UTI	Yes	7800
29.3	Female/8 y	*Achromobacter xylosoxidans*	HR-ALL, relapse	CRBSI	No	2100
37.4	Female/7 y	*Pseudomonas putida*	HR-ALL, relapse	CRBSI	Yes	0
41.4	Male/6 y	*Campylobacter jejuni*	HR-ALL, C/I	Unknown	Yes	1500
57.0	Male/5 y	*Yersinia enterocolitica*	HR-ALL, induction	CRBSI	No	100

ANC: absolute neutrophil count; C/I: Consolidation/intensification; CRBSI: Catheter-related bloodstream infection; HR-ALL: high-risk acute lymphoblastic leukaemia; UTI: urinary tract infection.

**Table 2 antibiotics-14-00847-t002:** Relationship between blood-culture time-to-positivity (TTP) and clinical and microbiological characteristics in immunocompromised paediatric patients with Gram-negative bacilli bloodstream infections.

	N (%)	Median TTP (p25–75)	*p*-Value
Total	128 (100)	11.4 (9.8–14.1)	
ANC (/mm^3^) at diagnosis			0.906
- <500	99 (77.3)	11.4 (10.1–13.4)	
- >500	29 (22.7)	8.7 (7.7–14.4)	
Previous antibiotic treatment			0.257
- Yes	20 (15.6)	11.5 (9.6–14.8)	
- No	108 (84.4)	11.3 (9.1–13.4)	
Disease-related risk factor			0.071
- ALL	38 (29.7)	11.4 (9.6–17.0)	
- AML	17 (13.3)	11.3 (8.9–12.0)	
- Lymphoma	5 (3.9)	13.6 (11.4–18.0)	
- Solid tumours	40 (31.3)	10.9 (8.8–12.3)	
- HSCT	20 (15.6)	11.8 (8.8–14.3)	
- Others	8 (6.2)	10.9 (8.4–13.6)	
Infectious focus			0.868
- Unknown	84 (65.6)	11.3 (10.0–13.0)	
- CRBSI	24 (18.8)	8.8 (7.5–13.9)	
- Complicated abdominal focus	13 (10.2)	12.7 (11.3–15.0)	
- Others	7 (5.4)	9.4 (8.1–17.6)	
Microorganism			<0.001
- *Escherichia coli*	41 (32.0)	10.5 (9.3–11.3)	
- *Klebsiella pneumoniae*	22 (17.2)	11.5 (8.6–12.0)	
- *Pseudomonas aeruginosa*	25 (19.5)	16.0 (13.9–16.8)	
- *Enterobacter cloacae*	12 (9.4)	10.2 (9.0–11.2)	
- Others	28 (21.9)	12.6 (8.0–19.5)	
Admission to the PICU			0.175
- Yes	15 (11.7)	10.5 (7.5–12.0)	
- No	113 (88.3)	11.3 (9.6–14.3)	
30-day mortality			0.236
- Yes	7 (5.5)	10.6 (7.9–11.3)	
- No	121 (94.5)	11.4 (9.4–14.3)	

ALL: high-risk acute lymphoblastic leukaemia; AML: acute myeloblastic leukaemia; ANC: absolute neutrophil count; CRBSI: Catheter-related bloodstream infection; HSCT: Hematopoietic stem cell transplantation; PICU: paediatric intensive care unit.

**Table 3 antibiotics-14-00847-t003:** Comparison of clinical characteristics of paediatric immunocompromised patients with GNB-BSI and blood-culture time-to-positivity (TTP) > 24 h or < 24 h.

	TTP < 24 h	TTP > 24 h	
	N (%)	N (%)	*p*-Value
Total	122 (95.3)	6 (4.7)	
Severe neutropenia (ANC < 500/mm^3^)	93 (76.2)	3 (50.0)	0.129
Previous antibiotic treatment	17 (13.9)	3 (50.0)	0.048
Disease-related risk factor			0.239
- ALL	33 (27.0)	5 (83.3)	
- AML	17 (13.9)	0 (0.0)	
- Lymphoma	5 (4.1)	0 (0.0)	
- Solid tumours	39 (32.0)	1 (16.7)	
- HSCT	20 (16.4)	0 (0.0)	
- Others	8 (6.5)	0 (0.0)	
Infectious focus			0.080
- Unknown	82 (62.7)	2 (33.3)	
- CRBSI	21 (17.2)	3 (50.0)	
- Complicated abdominal focus	13 (10.7)	0 (0.0)	
- Others	6 (4.9)	1 (16.7)	
Microorganism			0.005
- *Escherichia coli*	41 (33.6)	0 (0.0)	
- *Klebsiella pneumoniae*	21 (17.2)	1 (16.7)	
- *Pseudomonas aeruginosa*	25 (20.5)	0 (0.0)	
- *Enterobacter cloacae*	12 (9.8)	0 (0.0)	
- Others	23 (18.9)	5 (83.3)	
Admission to PICU	15 (12.3)	0 (0.0)	1.000
30-day mortality	7 (5.7)	0 (0.0)	1.000

ALL: high-risk acute lymphoblastic leukaemia; AML: acute myeloblastic leukaemia; ANC: absolute neutrophil count; CRBSI: catheter-related bloodstream infection; HSCT: hematopoietic stem cell transplantation; PICU: paediatric intensive care unit.

## Data Availability

The data presented in this study are available on request from the corresponding author upon request due to privacy reasons.

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
