# Peer review of "Blood Cultures Time-to-Positivity as an Antibiotic Stewardship Tool in Immunocompromised Children with Gram-Negative Bacteraemia"

_antibiotics, 2025, doi:10.3390/antibiotics14080847_

Round 1
Reviewer 1 Report
Comments and Suggestions for Authors
Introduction Section
From this section, the need for developing pediatric antimicrobial stewardship protocols/programs becomes clear. However, the following is not clear: the authors write in lines 96-103 that BC TTP can be effectively used in an antimicrobial stewardship program; what is the novelty of the manuscript then? BC TTP is not used for immunocompromised children?
The reviewer suggests shortening this section, leaving only those facts that would make it clear why the authors performed this study.
Materials and Methods Section. This section should be revised, since in its current form it is difficult to understand the study design. Perhaps structuring the methods that the authors used would help here. Maybe the study design could be depicted in the form of some kind of flow chart? There is almost no description of microbiological and other laboratory research methods here, it simply talks about the types of data that were collected.
Lines 321-339. This fragment does not contain information that is important for understanding the manuscript. Apparently the authors wanted to describe in detail the place where the study was performed; however, the reviewer believes that this part of the manuscript can be significantly shortened, and the manuscript will not suffer from it.
Line 358 (…antimicrobial agent with GN activity …); here, probably, antimicrobial agents active against gram-negative bacteria were meant. Should GN be replaced with GNB here? Because in other places in the manuscript GNB is everywhere, or the term GN pathogens occurs once (in line 400). The reviewer recommends to unify this.
Regarding Blood culture time-to-positivity in general. The authors write (lines 377-378) that the time period between blood sample collection and the start of incubation was not taken into account. The reviewer knows from the literature that TTP depends on many conditions: the number of bacteria present in the original blood sample; the type of microorganism; patient characteristics and laboratory practice, and other things. In this regard, a question arises. If some of the study conditions were not controlled, can the data be considered reliable? The reviewer understands that the method used by the authors provides an idea of ​​the severity of the infection, but this assessment must be greatly modified by the fact that the conditions for collecting and incubating samples are not standardized. It is unclear why the authors used catheter and peripheral cultures; the results section does not mention this. However, the reviewer knows from the literature that catheter cultures give an earlier positive result than peripheral cultures. Was there no difference?
The title of the manuscript (Time to positive blood culture as a tool for monitoring antibiotic therapy in immunocompromised children with gram-negative bacteremia) implies that the data obtained by the authors can be used to monitor antibiotic therapy. However, the materials and methods section does not provide any insight into how the authors achieved this. Moreover, the results section does not mention this either.
The results section also needs to be revised. This section is depressing; perhaps this is because the section describing the results is not always accompanied by graphic material, and the data that are in the tables are almost not described. In all cases, there is no interpretation of the results obtained, and this also makes it difficult to understand the manuscript. From all the content in this section, it is obvious that in the case of detection of gram-negative bacteria there is a small time-to-positivity; but this is also a known fact. The authors should explain why they did not exclude data on samples at the time of collection of which antibiotics against gram-negative bacteria were already used.
Author Response
We would like to sincerely thank you for taking the time to review our manuscript. Your comments were insightful and constructive, and they significantly contributed to improving the clarity and quality of the final version.
Please find our point-by-point responses to each of your comments below. All corresponding revisions have been made in the manuscript and are visible as tracked changes in the re-submitted files.
Introduction Section.
Comment 1: From this section, the need for developing paediatric antimicrobial stewardship protocols/programs becomes clear. However, the following is not clear: the authors write in lines 96-103 that BC-TTP can be effectively used in an antimicrobial stewardship program; what is the novelty of the manuscript then? BC TTP is not used for immunocompromised children?
Response 1: Thank you for pointing this out. Indeed, there is very little information published about BC-TTP in cancer and other paediatric immunocompromised patients. We have tried to improve the text and make this clearer. Now the sentence has been modified in the manuscript as follows (lines 112-116):
Nevertheless, paediatric evidence remains limited, consisting primarily of small, single-centre retrospective cohorts. To date, only one study has specifically examined children with FN, reporting that 86% of GNB-BSI BC became positive within 24 hours. However, this study included fewer than 20 events and was not specifically designed to evaluate TTP-guided antibiotic de-escalation [25].
Comment 2: The reviewer suggests shortening this section, leaving only those facts that would make it clear why the authors performed this study.
Response 2: Thank you for this suggestion. We have shortened the introduction and revised it to focus on the clinical problem, the knowledge gap, and the specific objective of our study.
Materials and Methods Section.
Comments 3: This section should be revised, since in its current form it is difficult to understand the study design. Perhaps structuring the methods that the authors used would help here. Maybe the study design could be depicted in the form of some kind of flow chart?
Response 3: Thank you for this helpful feedback. We have restructured the Materials and Methods section with the following sub-sections: 4.1. Study design and setting, 4.2. Study population, 4.3. Management of FN, 4.4. Microbiological procedures and definitions, 4.5. Data collection and outcomes, 4.6. Statistical analysis and 4.7. Ethical considerations; in order to improve clarity and readability.
Comments 4: There is almost no description of microbiological and other laboratory research methods here, it simply talks about the types of data that were collected.
Response 4: Thanks for pointing that out. We have also added a detailed description of the microbiological processing and blood culture incubation procedures. Now the sentence has been modified in the manuscript as follows (lines 370-382):
4.4. Microbiological procedures and definitions
BC are processed using the BACT/ALERT 3D automated microbial detection system (bioMérieux, Durham, NC, USA). When a BC flags positive, a Gram stain is immediately performed, followed by conventional subcultures on appropriate media (e.g., blood agar or other selective media, depending on the Gram stain result). Once bacterial growth is ob-served, microorganism identification is carried out using matrix-assisted laser desorption ionization–time of flight mass spectrometry (MALDI-TOF).
Antimicrobial susceptibility testing is performed using commercial panels, such as the MicroScan system (Beckman Coulter, USA). Mechanisms of multidrug resistance are further characterized using multiplex lateral-flow immunochromatographic assays, in-cluding the NG-Test CARBA 5 (NG Biotech, France) for the detection of carbapenemases and the NG-Test CTX-M MULTI (NG Biotech, France) for extended-spectrum β-lactamases (ESBLs).
Comment 5: Lines 321-339. This section includes excessive detail about the study site and can be shortened without impacting the manuscript.
Response 5: We agree and have shortened this section to retain only the information necessary for understanding the study setting and population characteristics. Now the sentence has been modified in the manuscript as follows (lines 332-337):
4.1. Study design and setting
We performed a retrospective observational single-centre cohort study in a tertiary paediatric hospital in Barcelona (Spain), including a dedicated centre for paediatric oncology with a capacity of 37 inpatient beds, 8 HSCT rooms, and 26 hospital day care units, where over 1,100 admissions of immunocompromised paediatric patients are managed every year.
Comment 6: Line 358 uses “GN,” while other parts of the manuscript use “GNB” or “GN pathogens.” The terminology should be unified.
Response 6: Thank you for pointing this out. We have reviewed the manuscript thoroughly and revised all terms to consistently use “GNB” (gram-negative bacilli) throughout.
Comment 7: Regarding Blood culture time-to-positivity in general. The authors write (lines 377-378) that the time period between blood sample collection and the start of incubation was not taken into account. The reviewer knows from the literature that TTP depends on many conditions: the number of bacteria present in the original blood sample; the type of microorganism; patient characteristics and laboratory practice, and other things. In this regard, a question arises. If some of the study conditions were not controlled, can the data be considered reliable?
Response 7: We appreciate this important point. While these factors were not standardized due to the retrospective nature of the study, our aim was to reflect real-world clinical practice. In our literature review most of the study's used this same methodology (e.g. Puerta-Alcalde P et al; Lambregts et al, or Dierig A et al). Nonetheless, we recognize this as a limitation, and this has been emphasized in the limitations section (lines 320-323):
We were unable to capture the time interval between sample collection and incubation in the Microbiology Laboratory, which led to an underestimation of the BC TTP that might have been clinically significant [34].
Comment 8: The authors included catheter and peripheral cultures but did not discuss this in the results. Catheter cultures are known to turn positive earlier than peripheral ones. Was there no difference?
Response 8: Thank you for highlighting this point. As clarified in the Methods section (lines 389-391), in patients with multiple blood cultures collected simultaneously (e.g., peripheral and central) at the onset of the episode, the shortest TTP was recorded. This approach was chosen because the earliest positive culture would represent the first microbiological result available to the clinical team and, therefore, the one most likely to influence early decision-making.
Only in cases of catheter-related bloodstream infections (CRABSI) does the blood culture drawn from the infected catheter become positive earlier than the concurrently obtained peripheral sample. This differential time to positivity is expected and required to meet the CRABSI definition. CRABSI accounted for 18% of all bloodstream infections included in our study. In the remaining 82%, no significant differences in time-to-positivity were observed between peripheral and central samples.
Comment 9: The title of the manuscript (Time to positive blood culture as a tool for monitoring antibiotic therapy in immunocompromised children with gram-negative bacteremia) implies that the data obtained by the authors can be used to monitor antibiotic therapy. However, the materials and methods section does not provide any insight into how the authors achieved this. Moreover, the results section does not mention this either.
Response 9: We thank the reviewer for this observation. The current title of the manuscript is “Blood culture time-to-positivity as an antibiotic stewardship tool in immunocompromised children with Gram-negative bacteraemia.”
The aim of our study was to evaluate whether TTP could be used as a practical tool to support antimicrobial stewardship, with a particular focus on the early de-escalation of broad-spectrum antibiotics in immunocompromised patients who are at high risk for multidrug-resistant (MDR) or severe infections. We believe the current title accurately reflects the scope and intent of our study.
Results section:
Comment 10: This section is depressing; perhaps this is because the section describing the results is not always accompanied by graphic material, and the data that are in the tables are almost not described. In all cases, there is no interpretation of the results obtained, and this also makes it difficult to understand the manuscript.
Response 10: We appreciate this helpful feedback. In response, we have revised the Results section and restructured it into the following sub-sections to improve clarity and readability: 2.1. Study Population; 2.2. Microbiological Findings; 2.3. Empirical Therapy and Clinical Outcomes; 2.4. BC-TTP Distribution and Associated Clinical and Microbiological Factors; 2.5. Factors Associated with TTP > 24 Hours.
Comment 11: The authors should explain why they did not exclude data on samples at the time of collection of which antibiotics against gram-negative bacteria were already used.
Response 11: Thank you for raising this important point. We included patients regardless of prior antibiotic exposure because our objective was to evaluate the utility of TTP in real-world clinical settings, where many immunocompromised children are already receiving empiric or prophylactic antibiotics at the time of suspected infection. In our study, 15.6% of patients were on antibiotics at the time of blood culture collection.
The potential impact of prior antibiotic use on TTP emerged as a key finding. As noted in the Discussion (lines 307-309), in patients with recent antibiotic exposure, our proposed TTP-based decision algorithm may not be applicable or reliable. This is further addressed in the Limitations section (lines 325-327), where we caution against extrapolating our findings to settings with different antibiotic policies, such as the routine use of fluoroquinolone prophylaxis.
Reviewer 2 Report
Comments and Suggestions for Authors
Major points
- Line 70 – “These antibiotic regimens carry a higher risk of drug-related adverse events, selection of MDR microorganisms, and lead to increased treatment costs.” – add some examples of specific drug-related adverse events; while the statement in itself is true, it requires references to support it
- Line 111 – “Nevertheless, data in immunocompromised children are scarce.” – are there no other studies that examine TTP in this patient population?
- Lines 115-116 – In my opinion the aim is overly simplified in this phrase and could be rephrased
- Materials and methods section should be restructured with several subsections – this aspect would improve the readability
- Materials and methods – “Local FN protocols included two aerobic BC (peripheral and central)” – somewhat unclear; does the case definition for FN in the local protocol require a total of two positive blood cultures from both peripheral and central line sources?
- Materials and methods – were anaerobic BC bottles not taken into consideration? There is only one mention of
- The Results section would benefit from improved organization to enhance clarity and readability – e.g., subsections; alternatively, tables or graphs could be added in addition to the text description to provide a more accessible overview of the data
- Results lines 142-158 – while this paragraph details the most commonly identified bacteria and some data regarding antimicrobial resistance, the methods section lacks any information regarding these aspects; add the relevant microbiological details regarding identification, method of antimicrobial susceptibility testing and
- Results Figure 2 – while the proposed algorithm seems sensible, I believe it must be categorically stated that the relatively low number of cases included represent a limitation and that further larger sample sizes are required to validate such an algorithm
Minor points
- line 78 – length of hospitalisation perhaps?
- line 89 – “International Clinical Trials Registry Platform included children)” – change font to black
- line 97 "is a useful and usually accessible data” – rephrase
- line 370 “andits phenotypic susceptibility” – and its
Author Response
We would like to express our sincere thanks for your thorough and thoughtful review of our manuscript. Your feedback was highly valuable and helped us enhance the clarity, structure, and overall quality of the work.
Below, we provide detailed responses to each of your comments. All corresponding changes have been incorporated into the revised manuscript and are highlighted using tracked changes in the re-submitted documents.
MAJOR POINTS
Comment 1: Line 70 – “These antibiotic regimens carry a higher risk of drug-related adverse events, selection of MDR microorganisms, and lead to increased treatment costs.” – add some examples of specific drug-related adverse events; while the statement in itself is true, it requires references to support it.
Response 1: Thank you for pointing this out. We agree with this comment. Therefore, we have updated the sentence in the manuscript and have included some references, as follows (lines 69-73):
These antibiotic regimens are associated with important adverse events, such as nephro- and ototoxicity with aminoglycosides, neurotoxicity and seizures with carbapenems, or Clostridioides difficile infection and gut microbiota disruption with broad-spectrum β-lactams. They also promote the selection of MDR microorganisms, and substantially in-crease treatment costs [4, 10, 11].
Comment 2: Line 111 – “Nevertheless, data in immunocompromised children are scarce.” – are there no other studies that examine TTP in this patient population?
Response 2: Thank you again. Indeed, there is very little information published in TTP in cancer and other immunocompromised patients. We have tried to improve this sentence and make this clearer. The new paragraph reads (lines 112-116):
Nevertheless, paediatric evidence remains limited, consisting primarily of small, sin-gle-centre retrospective cohorts. To date, only one study has specifically examined children with FN, reporting that 86% of GNB-BSI BC became positive within 24 hours. However, this study included fewer than 20 events and was not specifically designed to evaluate TTP-guided antibiotic de-escalation [25].
Comment 3: Lines 115-116 – In my opinion the aim is overly simplified in this phrase and could be rephrased.
Response 3: Thank you. The sentence has been rephrased as suggested in the manuscript, and this is the updated version (lines 119-123):
The primary objectives of this study were to characterise the distribution of BC-TTP among immunocompromised children and adolescents with GNB-BSI and to investigate patient-, clinical- and pathogen-related factors that influence TTP, with the ultimate goal of informing evidence-based, early antibiotic de-escalation strategies in this high-risk popu-lation.
Comment 4: Materials and methods section should be restructured with several subsections – this aspect would improve the readability.
Response 4: Thank you. We agree with the suggestion. We have, accordingly, re-structured the section in the manuscript, with the following sub-sections: 4.1. Study design and setting, 4.2. Study population, 4.3. Management of FN, 4.4. Microbiological procedures and definitions, 4.5. Data collection and outcomes, 4.6. Statistical analysis and 4.7. Ethical considerations.
Comment 5: Materials and methods – “Local FN protocols included two aerobic BC (peripheral and central)” – somewhat unclear; does the case definition for FN in the local protocol require a total of two positive blood cultures from both peripheral and central line sources?
Response 5: Thank you for pointing out that the information was not clear enough. We have updated this in the manuscript as follows (lines 355-359):
4.3. Management of FN
According to our local FN protocol, at the onset of every febrile episode two aerobic blood-culture sets are obtained before the first dose of empirical antibiotics: one from a pe-ripheral vein and one from the central venous catheter (or from each lumen if multi-lumen). Anaerobic bottles are not collected routinely.
Comment 6: Materials and methods – were anaerobic BC bottles not taken into consideration? There is only one mention of
Response 6: Thank you for pointing out that the information was not clear enough. We do not routinely collect anaerobic blood cultures, and this has now been clarified (line 359).
Comment 7: The Results section would benefit from improved organization to enhance clarity and readability – e.g., subsections; alternatively, tables or graphs could be added in addition to the text description to provide a more accessible overview of the data.
Response 7: Thank you. We agree, again, with the suggestion. To improve clarity and readability, we have added the following sub-sections: 2.1. Study Population; 2.2. Microbiological Findings; 2.3. Empirical Therapy and Clinical Outcomes; 2.4. BC-TTP Distribution and Associated Clinical and Microbiological Factors; 2.5. Factors Associated with TTP > 24 Hours.
Comment 8: Results lines 142-158 – while this paragraph details the most commonly identified bacteria and some data regarding antimicrobial resistance, the methods section lacks any information regarding these aspects; add the relevant microbiological details regarding identification, method of antimicrobial susceptibility testing and.
Response 8: Thank you. Indeed, microbiological methodology was not enough detailed in the actual version of the submitted manuscript. We have added a detailed description of the microbiological processing and blood culture incubation procedures (lines 370-382):
4.4. Microbiological procedures and definitions
BC are processed using the BACT/ALERT 3D automated microbial detection system (bioMérieux, Durham, NC, USA). When a BC flags positive, a Gram stain is immediately performed, followed by conventional subcultures on appropriate media (e.g., blood agar or other selective media, depending on the Gram stain result). Once bacterial growth is ob-served, microorganism identification is carried out using matrix-assisted laser desorption ionization–time of flight mass spectrometry (MALDI-TOF).
Antimicrobial susceptibility testing is performed using commercial panels, such as the MicroScan system (Beckman Coulter, USA). Mechanisms of multidrug resistance are further characterized using multiplex lateral-flow immunochromatographic assays, in-cluding the NG-Test CARBA 5 (NG Biotech, France) for the detection of carbapenemases and the NG-Test CTX-M MULTI (NG Biotech, France) for extended-spectrum β-lactamases (ESBLs).
Comment 9: - Results Figure 2 – while the proposed algorithm seems sensible, I believe it must be categorically stated that the relatively low number of cases included represent a limitation and that further larger sample sizes are required to validate such an algorithm.
Response 9: Thank you. Indeed, our intention is to present a simplified clinical algorithm that serves as a preliminary model; however, we acknowledge that its implementation requires formal validation. To clarify this important point, we have added the following sentence (lines 296-298):
It should be kept in mind that this preliminary proposal is exploratory in nature and requires validation in external cohorts, ideally through larger, prospective, multicenter studies.
MINOR POINTS
Comment 10: - line 78 – length of hospitalization perhaps?
Response 10: The word “stay” has been updated in the manuscript as “hospitalization”.
Comment 11: line 89 – “International Clinical Trials Registry Platform included children)” – change font to black
Response 11: Thank you. Line 89 has been changed to black font in the manuscript.
Comment 12: - line 97 "is a useful and usually accessible data” – rephrase
Response 12: Thank you for the suggestion. We agree and, accordingly, we have rephrased the sentence in the manuscript as follows (lines 97-98):
Blood-culture (BC) time-to-positivity (TTP)—the interval from the start of incubation to a positive alarm—is a readily available and clinically informative metric.
Comment 13: - line 370 “andits phenotypic susceptibility” – and its
Response 13: Thank you. We have included a space between the word “and” and the word “its” in the updated version of the manuscript.
Round 2
Reviewer 1 Report
Comments and Suggestions for Authors
The authors have fully responded to all of the reviewer's questions and comments; the quality of the revised manuscript is high. The reviewer believes that the manuscript can be accepted for publication.
Author Response
We thank the Reviewer for the kind words.
Reviewer 2 Report
Comments and Suggestions for Authors
Thank you for the performed changes to the manuscript.
Author Response
We thank the Reviewer for his/her kind words.